The micro-niche explains allotopy and syntopy in South American Liolaemus (Iguania: Liolaemidae) lizards

Quinteros Andrés S. 1 2 sebasquint@gmail.com
http://orcid.org/0000-0002-1024-5568 Portelli Sabrina N. 1 3
1 Instituto de Bio y Geociencias del NOA—IBIGEO—UNSa—CONICET , Salta , Argentina
2 Facultad de Ciencias Naturales—Universidad Nacional de Salta, Cátedra de Sistemática Filogenética , Salta , Argentina
3 Facultad de Ciencias Naturales—Universidad Nacional de Salta, Catedra de Zoología , Salta , Argentina
Santamaria Carlos
Electronic publication date: 2025 Feb 17
Publication date: 2025
Volume: 13
Electronic Location ID: e18979
Received 2024 Aug 22; Accepted 2025 Jan 22
Copyright: © 2025 Quinteros and Portelli
Copyright year: 2025
Copyright holder: Quinteros and Portelli
License: This is an open access article distributed under the terms of the Creative Commons Attribution License, which permits unrestricted use, distribution, reproduction and adaptation in any medium and for any purpose provided that it is properly attributed. For attribution, the original author(s), title, publication source (PeerJ) and either DOI or URL of the article must be cited.
License URL: https://creativecommons.org/licenses/by/4.0/

Keywords: Argentina, Species distribution, Ectotherms, Allotopy, Syntopy

Funding: FONCYT PICT‐2021‐GRFTI‐00186 PIP CONICET-1120200100954 and CIUNSa N° 2931/0 PICT-2021-I-INVI-00715 This work was supported by FONCYT PICT‐2021‐GRFTI‐00186; PIP CONICET-1120200100954 and CIUNSa N° 2931/0, from Sebastián Quinteros and by PICT-2021-I-INVI-00715 from Sabrina Portelli. The funders had no role in study design, data collection and analysis, decision to publish, or preparation of the manuscript.

==============================
Species distribution models have been established as essential tools for projecting the effects of changing environmental conditions on species distribution across space and time. The microclimatic niche denotes the environmental conditions within a habitat at a small scale or localized area. These conditions have a direct influence on several ecological traits and on species distribution as these conditions determine which organisms can survive and/or reproduce. This study examines the microclimate data from four sites located in Northwestern Salta Province, Argentina. Four South American Liolaemus lizard species were found to inhabit these four sites in allotopy or syntopy, with Liolaemus irregularis inhabiting all four sites. Liolaemus irregularis is the sole Liolaemus species inhabiting Site 1; L. irregularis inhabits Site 2 in syntopy with L. multicolor; L. irregularis inhabits Site 3 in syntopy with L. yanalcu; and L. irregularis inhabits Site 4 in syntopy with L. albiceps. To characterize the four sites, a microclimate model was generated for an interval from 10 AM to 6 PM every day, for 10 years. The sites exhibited some differences in the combination of climatic and soil characteristics. Site 1 was characterized by low relative humidity, high temperature, high wind speed, and Cambisol soil type. Site 2 had high relative humidity, low temperature, moderate wind speed, and Andosol soil type. Site 3 had high relative humidity, high temperature, low wind speed, and Cambisol soil type. Site 4 had high relative humidity, low temperature, moderate wind speed, and Regosol soil type. Temperature, humidity, wind speed, soil type, and species diet influenced the presence of lizard species at each site. It is evident that microenvironmental conditions profoundly influence lizard distribution and biological interactions.

Introduction

Species distribution models (SDMs) have become essential tools for projecting the effects of changing environmental conditions on species distribution across space and time (Guisan & Zimmermann, 2000; Guisan & Thuiller, 2005). Most datasets used in macroclimate analyses are based on interpolations from standardized weather station data, typically using temperature measurements taken outside forests and over grasslands at approximately 2 m above ground level (De Frenne et al., 2021). While these data are adequate for capturing changes in open-air temperatures, issues arise when using them to model the responses of species living close to the ground in topographically heterogeneous terrains and/or ecosystems with trees and shrubs. Because of this, appropriately scaled microclimatic data must be integrated into SDMs and ecological research more broadly (Körner & Hiltbrunner, 2018).

Several studies have acknowledged the utility of niche modeling, particularly for ectotherms and species inhabiting seasonal environments (Lembrechts, Nijs & Lenoir, 2019; Caetano et al., 2020; Brodie et al., 2021; Pérez Navarro et al., 2021; Toro-Cardona, Parra & Rojas-Soto, 2023). Recent advances in microclimate mapping have been driven by sophisticated microclimate measurement and modeling techniques (Maclean, 2019; Zellweger et al., 2019; Maclean et al., 2021). The microclimatic niche refers to the environmental conditions (temperature, relative humidity, solar radiation, wind speed, etc.) within a habitat at a small scale or localized area. These conditions directly influence the ecology, diversity, morphology, physiology, and distribution of species (Jiménez-Robles & De la Riva, 2019; Vives-Ingla et al., 2023; Beugnon et al., 2024; König et al., 2023; Stickley & Fraterrigo, 2023; Toro-Cardona, Parra & Rojas-Soto, 2023; Kemppinen et al., 2024), as these conditions determine which organisms can survive and/or reproduce (Bramer et al., 2018). To accurately assess the effects of microclimate conditions on individual animals, it is crucial to ensure that the spatial scale of the climate data aligns with the behavior and physiology of the organisms of interest (Ashcroft, 2010; Dobrowski, 2011; Lembrechts, Nijs & Lenoir, 2019).

Research to date has primarily focused on short-term species distributions, limiting its utility for testing the extent to which population trends are affected by microclimate conditions, especially when microclimatic variations play a crucial role in the effect of climate change on organisms (Dobrowski et al., 2015). Researchers and conservationists must understand the heterogeneity of microclimates and the susceptibility of species to climate change in order to mitigate the effect of these changes (Morelli et al., 2016). Given their sensitivity to environmental changes and thermal dependence, reptiles are of particular interest for studying niche and microhabitat selection in regions with different habitats and climates (Prieto-Ramirez et al., 2018).

Studies have demonstrated that ectotherms can be affected by changes in humidity (Bodineau et al., 2024) or wind (Ortega, Mencía & Pérez-Mellado, 2017; Spears et al., 2024), which can influence habitat selection, as observed in lizard species. Recent studies have investigated the microclimatic niches of lizards, with most studies focusing on thermoregulation (Ortega, Mencía & Pérez-Mellado, 2017; Jiménez-Robles & De la Riva, 2019; Moore, Stow & Kearney, 2018; Toro-Cardona, Parra & Rojas-Soto, 2023; Dufour et al., 2024) or species distribution (Schmitz et al., 2022; Kang et al., 2024). Studies on community structure based on the microclimatic niches of lizards are more limited (Costa et al., 2020; Souza-Oliveira et al., 2024). Moreover, the studies on community structure that use microclimate data typically examine community structure in heterogeneous environments. It is well established that spatial heterogeneity, influenced by vegetation structure affecting factors such as light, temperature, and humidity, plays a crucial role in species distribution and community structure (Oliveira, Rocha & Bagnall, 1994; Cerqueira et al., 2003; Radder, Saidapur & Shanbhag, 2005; Dias & Rocha, 2014; Nunes Cosendey, Duarte da Rocha & de Menezes, 2019). So far, no exhaustive analysis has yet been conducted on different microclimatic niches that includes multiple closely-distributed species sharing the same habitat.

Liolaemus is one of the most speciose lizard genera in the world, encompassing more than 270 species (Abdala et al., 2023). Liolaemus species have a widespread distribution in South America, ranging from southern Tierra del Fuego in Argentina to central Peru (Abdala et al., 2023; Quinteros, Ruíz-Monachesi & Abdala, 2020). Phylogenetically, the genus is divided into two subgenera: Liolaemus sensu stricto and Eulaemus (Laurent, 1983, 1985; Schulte et al., 2000; Espinoza, Wiens & Tracy, 2004; Lobo, Espinoza & Quinteros, 2010; Abdala & Quinteros, 2014; Panzera et al., 2017; Esquerré et al., 2019). Additionally, various sections, series, and groups have been proposed within these subgenera over the years (Ortiz, 1981; Cei, 1986, 1993; Etheridge, 1995; Schulte et al., 2000; Lobo, 2001, 2005; Abdala, 2007; Quinteros, 2012; Abdala & Quinteros, 2014; Troncoso-Palacios et al., 2015; Portelli & Quinteros, 2018; Abdala et al., 2020; Quinteros, Ruíz-Monachesi & Abdala, 2020; Portelli et al., 2022; Abdala et al., 2023; among others). Due to the high diversity of Liolaemus, it is possible to find species from different clades cohabiting in certain areas. For instance, a northern locality in the Argentine Andes hosts four species: Liolaemus albiceps, L. irregularis, Liolaemus multicolor, and L. yanalcu. A study on the thermoregulation efficiency of these species by Valdecantos et al. (2013) identified four closely situated sites where these species coexist. In one site, L. irregularis inhabits allotopycally, while in the other three sites, L. irregularis coexists syntopycally with one of the other three species. These four species belong to different phylogenetic clades within the Liolaemus genus. Liolaemus albiceps, L. irregularis, and L. multicolor are members of the Eulaemus subgenus. Within this subgenus, L. multicolor is a member of the L. montanus section, while L. albiceps and L. irregularis are members of the L. boulengeri section. Moreover, L. albiceps and L. irregularis are sister taxa (Abdala, 2007). Liolaemus yanalcu is the most phylogenetically distant taxon, as it is a member of the Liolaemus sensu stricto subgenus. These four species have both unique and shared biological traits. Liolaemus irregularis is an omnivorous species, while the other three species are herbivores (Abdala et al., 2021); Liolaemus yanalcu is oviparous, while the other three species are viviparous (Martinez Oliver & Lobo, 2002; Abdala, 2007, 2021). Liolaemus albiceps, L. irregularis, and L. multicolor use shrubs (Adesmia sp.) as shelter, whereas L. yanalcu uses grassland (Festuca sp.). These four species also differ in size. L. albiceps and L. irregularis are larger species and share a similar snout-vent-length (SVL), L. multicolor is medium sized, and L. yanalcu is the shortest of the four species (Abdala, 2007; Valdecantos & Lobo, 2007; Quinteros, 2012).

The aim of this study was to determine the possible factors that shape the assembling patterns (allotopy or syntopy) of four Liolaemus species (L. albiceps, L. irregularis, L. multicolor, and L. yanalcu) in northwestern Argentina, with a focus on both microenvironmental variables and intrinsic biological characteristics. To achieve this, we characterized and compared the microclimatic variables and soil types across four sites to assess environmental variability. Furthermore, we analyzed the relationship between the presence or absence of these species in those sites and both microenvironmental variables and biological traits. Additionally, we investigated the associations between microclimatic variables and biological characteristics of the species, such as parity mode, diet, and thermoregulatory abilities. Lastly, we explored the potential influence of phylogenetic relationships on the spatial distribution and assembly patterns of these species. Based on these objectives, this study will test the following hypotheses based on the study sites being close together geographically, having homogeneous landscapes, and hosting different species compositions: 1) Microenvironmental variables could act as a heterogeneous mosaic, shaping different species compositions across sites than large-scale landscape environmental variables; 2) Since organisms tend to reflect the conditions of their environment, a relationship is expected between various biological traits and the microclimatic variables studied; 3) If species exhibit a distributional pattern driven by microclimatic variables, then phylogenetic relationships among them have not influenced their distribution across sites.

Materials and Methods

Species and study area

The microclimate data from four sites located near San Antonio de Los Cobres, in northwestern Salta Province, Argentina were analyzed (Fig. 1). These sites were previously identified by Valdecantos et al. (2013). Liolaemus irregularis is the sole Liolaemus species inhabiting Site 1; L. irregularis inhabits Site 2 in syntopy with L. multicolor; L. irregularis inhabits Site 3 in syntopy with L. yanalcu; and L. irregularis inhabits Site 4 in syntopy with L. albiceps. These sites were selected for studying the microclimate niche because of the following: (1) their close proximity (less than 30 km maximum distance); (2) the homogeneous landscape across the four sites; and (3) the lizard species composition at each site. The study area is characterized by shrubs ranging from 0.5–1 m tall, such as Adesmia horrida, Baccharis incarum, Fabiana densa, and Parastrephia quadrangularis (Cabrera, 1976). However, Site 3 also features a grassland composed of Festuca sp.

Figure 1 Map showing the study area and the location of the four sites.

(Note the close distribution of the sites).

Microclimatic niche

To characterize the four sites, a microclimate model was generated at hourly intervals from 10 AM to 6 PM every day over a period of 10 years (2013–2023), using the NicheMapR package (Kearney & Porter, 2017), implemented in R 4.4.0 (R Core Team, 2024). These hourly intervals corresponded to the daily activity patterns of the studied species (Villavicencio et al., 2001; Castillo et al., 2015). NicheMapR uses global weather data provided by the Climate Research Unit to create microclimate models with high temporal resolution. The microclimate of each site was modeled for air temperature (°C), sun radiation (W/m2), relative humidity (%), and wind speed (m/s) at a height of 1 cm. Each occurrence in the microclimate model was characterized by extracting elevation, slope, aspect, minimum and maximum shade, and soil type from various sources. Elevation data were obtained from a digital elevation model (DEM) from the EarthEnv database (Amatulli et al., 2018; https://www.earthenv.org/) at a 90 m resolution; slope and aspect were calculated from the DEM using the R package raster (Hijmans & van Etten, 2012). Soil data were extracted from the SoilGrids database (Hengl et al., 2017; https://soilgrids.org/) using classifications from the World Reference Base for Soil Resources (WRB). Since the WRB does not have a direct equivalent to the soil type configuration of NicheMapR (which corresponds to the soil texture triangle, United States Department of Agriculture (USDA), 1987), each group was reclassified based on its general composition and physical characteristics (ISRIC, 2019). To obtain the shade values, the tree cover layer from 2000 (Hansen et al., 2013) was used. After collecting all the information for each occurrence, a microclimate model was generated on an hourly basis for each day of the year (365 days) over the 10-year period at ground level (1 cm). The data were then pruned to keep only those corresponding to the activity period of the lizards, between 10 AM and 6 PM (See Supplemental File 1 for code details).

Statistical analyses

A discriminant function analysis (DFA) was performed to summarize the differences between the four sites. A repeated measures ANOVA was then performed to elucidate the differences of each variable between the study sites (Weinfurt, 2000). Assumptions such as normality (Shapiro-Wilk normality test), sphericity (Mauchly’s test), and the absence of significant outliers were also checked. Post hoc analyses were conducted using the Bonferronni adjustment (Emerson, 2020). Analyses were performed in R 4.4.0 using the tidyverse, ggpurb, and rstatix packages. Monthly comparisons were made for each variable. The annual variation of air temperature, relative humidity, wind speed, and sun radiation was then plotted. These plots were generated using R 4.4.0 (R Core Team, 2024) with the ggplot2 package.

A multiple logistic regression was then performed to identify possible correlations between microclimatic niche variables, biological traits, and the presence/absence of the species in the four sites. A stepwise regression, using the Akaike information criterion (AIC), was performed to identify the best predictors.

Linear regression models were employed to examine the relationships between the microclimatic data and the biological traits of the species. Temperature data were taken from Valdecantos et al. (2013).

A phylogenetic mixed model was used to explore the influence of the phylogenetic relationships of the species on the distribution of the species. Analyses were performed in R 4.4.0 using the ape, broom, car, MCMCglmm, phytools, pscl, and tidyverse packages.

Results

The discriminant analysis showed that Site 1 was different from the others, but was the most similar to Site 3. Sites 2 and 4 had similar values (Fig. 2). Site 1 was characterized by low relative humidity, high temperature, high wind speed, and Cambisol soil type. Site 2 had high relative humidity, low temperature, moderate wind speed, and Andosol soil type. Site 3 had high relative humidity, high temperature, low wind speed, and Cambisol soil type. Site 4 had high relative humidity, low temperature, moderate wind speed, and Regosol soil type.

Figure 2 Discriminant function analysis (DFA) summarizing differences between the four sites.

Though the landscapes of the four sites are homogeneous, differences in microclimatic niches among the sites were evident throughout the year (Fig. S1). However, there were no significant differences in solar radiation between the sites. Site 1 was the most distinct site, but shared similarities with Site 3 in terms of air temperature and soil type. Conversely, Sites 2 and 4 had similar microclimatic values, differing primarily in soil type.

Site 3 had the highest temperatures and Site 4 recorded the lowest temperatures. Temperatures were similar at Sites 1 and 3 and at Sites 2 and 4, but temperatures differed between these pairs (Table 1). Significant differences were evident during the months when lizard activity peaked (November to February; Fig. S2). Site 1 consistently had the lowest relative humidity throughout the year (Fig. S1), and there was no significant difference in humidity between the other three sites. Wind speed was highest at Site 1 throughout the year, and Site 3 recorded the lowest wind speed values. Sites 2 and 4 had overlapping wind speed values throughout the year, with variations occurring only in May.

Table 1 Monthly average and standard deviation of the variables studied by site.

SITE	JAN	FEB	MAR	APR	MAY	JUN	JUL	AUG	SEP	OCT	NOV	DEC	
TEMP	
1	16.4	15.8	14.9	13.1	10.7	8.12	8.34	10.2	12.5	15.3	16.6	16.4	
0.19	0.16	0.49	0.55	0.89	0.45	0.43	0.61	0.85	0.63	0.16	0.16	
2	15.3	14.6	13.8	11.9	9.55	6.87	7.02	8.98	11.3	14.2	15.5	15.3	
0.20	0.16	0.47	0.58	0.85	0.45	0.41	0.64	0.82	0.63	0.15	0.18	
3	16.9	16.2	15.2	13.4	10.9	8.37	8.52	10.5	12.9	15.7	17.1	16.9	
0.21	0.18	0.71	0.59	0.88	0.44	0.44	0.65	0.88	0.64	0.15	0.18	
4	15.1	14.4	13.5	11.7	9.13	6.6	6.78	8.71	11.1	13.9	15.2	15.1	
0.21	0.17	0.47	0.58	1.14	0.45	0.44	0.64	0.88	0.63	0.15	0.18	
HUM	
1	36.9	38.5	36.5	30.1	24.2	21.1	20.5	19.9	21.6	24.8	29.2	36.9	
1.65	0.19	1.24	2.18	1.11	0.42	0.46	0.17	0.85	0.89	2.12	1.61	
2	44.7	46.7	44.3	37.0	30.2	27.9	25.7	24.5	26.2	30.0	35.3	44.7	
2.01	0.25	1.45	2.5	1.23	0.51	0.63	0.13	0.89	1.11	2.57	1.99	
3	44.1	45.9	44.2	36.8	30.1	27.8	25.5	24.4	25.9	29.8	34.9	44.1	
1.97	0.23	2.65	2.46	1.23	0.52	0.65	0.09	0.94	1.08	2.5	1.95	
4	45.4	47.4	45.0	37.6	31.2	28.4	26.1	25.0	26.7	30.5	35.9	45.4	
2.03	0.25	1.45	2.53	1.34	0.51	0.67	0.19	0.96	1.09	2.6	2.03	
WIND	
1	2.16	2.08	2.02	1.94	1.9	2.04	2.07	2.2	2.36	2.34	2.32	2.16	
0.04	0.01	0.02	0.03	0.03	0.03	0.01	0.06	0.02	0.01	0.03	0.04	
2	2.04	1.96	1.9	1.82	1.78	1.92	1.95	2.08	2.24	2.22	2.25	2.04	
0.06	0.01	0.02	0.03	0.03	0.03	0.01	0.06	0.02	0.02	0.03	0.06	
3	1.92	1.84	1.79	1.7	1.66	1.8	1.83	1.95	2.12	2.15	2.13	1.92	
0.05	0.01	0.09	0.03	0.03	0.03	0.01	0.06	0.02	0.01	0.04	0.05	
4	2.04	1.96	1.9	1.82	1.77	1.92	1.95	2.08	2.24	2.22	2.25	2.04	
0.06	0.01	0.02	0.03	0.03	0.03	0.01	0.06	0.02	0.01	0.03	0.06	
RAD	
1	559.26	539.23	541.38	493.83	449.06	410.16	441.28	510.55	580.21	618.08	620.32	563.25	
161.03	163.73	182.91	190.35	192.31	186.23	195.66	206.47	211.31	195.17	180.02	158.11	
2	559.29	533.77	540.99	492.71	447.61	409.80	441.64	511.15	583.23	618.33	619.24	563.65	
158.80	162.92	182.72	189.85	191.56	185.93	195.68	206.58	208.53	195.20	179.67	156.42	
3	555.82	535.60	540.39	493.18	448.75	410.77	442.79	511.01	581.88	617.53	617.36	559.43	
158.48	163.09	182.79	190.18	191.88	185.70	195.94	206.53	208.29	195.42	179.78	155.80	
4	560.11	536.68	538.74	492.43	447.19	409.37	441.17	510.82	583.08	618.35	619.41	563.68	
159.07	162.86	185.48	189.79	191.47	185.88	195.61	206.53	208.47	195.12	179.60	156.65	
Note:

TEMP, Air Temperature; HUM, Realtive Humidity; WIND, Wind speed; RAD, Sun Radiation. Values correspond to mean (above) and standard deviation (below).

The repeated measure ANOVA of monthly variables across sites (Table S1) showed significant differences in all studied variables except for solar radiation. Temperatures varied significantly throughout the year between Sites 1 and 3 and between Sites 2 and 4. There was a significant difference in relative humidity in all months between Site 1 and Sites 2, 3, and 4. Wind speeds were significantly different between Site 1 and Sites 2, 3, and 4, and between Site 3 and Sites 2 and 4, with no significant differences found between Sites 2 and 4. Soil types varied across sites, with Sites 1 and 3 characterized by Cambisol soil type, Site 2 characterized by Andosol soil type, and Site 4 characterized by Regosol soil type (Table 2).

Table 2 Soil type for each site.

Based on Soilgrid.

SITE	WRB group	Soil type classification following ISRIC	NicheMapR soil type code	
SITE 1	Cambisols	Sandy clay	9	
SITE 3	
SITE 2	Andosols	Silt	5	
SITE 4	Regosols	Silty clay	10	

The stepwise regression discarded radiation data, so the multiple logistic regression of the microclimatic niche variables was performed using temperature, humidity, and wind speed. The results showed that temperature and humidity effected the distribution of the species, whereas wind speed had a low influence on species distribution. When temperature and wind speed increased, the probability of the presence of the species decreased (estimate = −0.300, p < 0.01; estimate = −0.050, p = 0.04; respectively). Conversely, when humidity increased, the probability of the presence of the species increased (estimate = 0.200, p < 0.01). The Andosol soil type also increased the probability of the presence of the species (estimate: 1.5; p < 0.05).

There was a relationship between species diet and the presence of the species (estimate: 19.66; p = 0.01), with omnivory increasing the probability of species presence at the sites. Parity mode was not related to the presence of the species at the sites.

The results of the analysis of relationships between microclimatic niche variables and biological traits showed that body temperature was related to both temperature and humidity (p < 0.01 and p = 0.03; R2 = 0.85); Body temperature increased when temperature and humidity increased. Preferred temperature was only related to temperature (p < 0.01; R2 = 0.90). There was no relationship between phylogeny and the presence of the species at the sites.

Discussion

The species included in this study and their distribution across four sites, forming pairs of species, were first examined by Valdecantos et al. (2013) within a thermoregulation framework. They found that these species are efficient thermoregulators, similar to many other Liolaemus species (Carothers et al., 1997; Carothers, Marquet & Jaksic, 1998). However, they did not find evidence to support thermal niche segregation, as proposed by Pianka & Huey (1978), since the species showed similar body temperatures measured in the field. The present study is the first to use microclimatic data to elucidate the community structure of Liolaemus species. The findings of this study underscore the critical role of microclimate variability in shaping species distributions. Our results support our hypotheses, demonstrating that microenvironmental variables shape distinct species compositions across sites, are associated with certain biological traits of the species, and show no relationship with phylogeny.

Despite the homogeneity of the landscape across the four study sites, which primarily reflects the shared vegetation type, there were recorded differences in air temperature, relative humidity, wind speed, and soil type between the sites. Liolaemus irregularis appears to be the most adaptable species of the four Liolaemus species in this study, inhabiting all four sites. In contrast, L. yanalcu requires an environment with high humidity and temperature, low wind speed, and a soil type that retains both heat and humidity (Cambisol). Liolaemus albiceps and L. multicolor occupy sites with similar characteristics of high humidity, low temperature, and moderate wind speed. These two species differ in their soil type use, with L. multicolor favoring a soil type that retains humidity and heat (Andosols) and L. albiceps preferring a soil type with poor humidity and heat retention (Regosols).

The wide distribution of Liolaemus irregularis and the findings of this study both indicate that this species exhibits a generalist profile, inhabiting sites with diverse temperature, humidity, wind speed, and soil types. Temperature is known to be a crucial variable shaping reptile species distribution (Meiri et al., 2013). The results support our first hypothesis as this study found that higher temperatures decreased the probability of the presence of Liolaemus species. Despite Valdecantos et al.’s (2013) report of no statistical differences among the four species, Liolaemus irregularis exhibits the widest range of body temperatures (28.3–39.5 °C), which helps explain why this was the only Liolaemus species found at Site 1. The results from Site 3, where Liolaemus yanalcu is able to regulate its body temperature and minimize its evaporative water loss due to higher temperatures and high relative humidity, highlight the importance of the interactions between thermal inertia and microclimatic conditions. Small-sized lizards, like L. yanalcu (62 mm max SVL), exhibit low thermal inertia, enabling rapid heat exchange with their environment, making them highly sensitive to thermal fluctuations (Herczeg, Török & Korsós, 2007; Sagonas et al., 2013; Moreno et al., 2016; Claunch et al., 2021). However, this same characteristic may enhance their thermoregulatory efficiency in warm and humid environments such as Site 3, where conditions facilitate a balance between thermal gain and reduced evaporative water loss. This balance reflects the concept of thermo-hydroregulation, which integrates physiological and behavioral processes to maintain both water and thermal balance (Rozen-Rechels et al., 2019). In this case, the high relative humidity at Site 3 likely reduces the evaporative cost associated with thermoregulation, thereby optimizing the performance and survival of L. yanalcu. These findings align with previous studies emphasizing how interactions among temperature, humidity, and landscape structure determine the availability of microhabitats for small ectotherms (Guillon et al., 2013; Sears et al., 2016) and how water, rather than temperature, may represent a more limiting factor under climate change scenarios (Kearney et al., 2018; Mi et al., 2022). Additionally, Site 3 features some Festuca sp. grasslands, which serve as shelter for L. yanalcu (Martinez Oliver & Lobo, 2002). This shelter likely serves two functions: providing protection against predators, as lizards inside the shelter are difficult to detect, and acting as a thermally favorable microhabitat, as individuals have been observed basking outside the grassland (C Abdala, S Quinteros, 2017–2020, personal observations). Vegetation and habitat structure can significantly influence shelter selection and, consequently, the survival of lizards (Smith & Ballinger, 2001). There exist many studies on lizards which support the use of shelters as protection against predators and for thermoregulation (Carrascal et al., 1992; Smith & Ballinger, 2001; Martín, López & Cooper, 2003; Herczeg et al., 2008; Monasterio et al., 2009), even for Liolaemus lizards (Fuentes & Cancino, 1979; Stellatelli et al., 2015, 2016). Further studies are required to confirm or refute these hypotheses in our focal species. Given the microclimatic conditions of Site 3, the grassland likely provides refuge from hydric stress, as relative humidity is higher within the grassland compared to the surface (Kearney & Porter, 2004; Treilibs et al., 2016; Sannolo & Carretero, 2019; Trewartha et al., 2024). Zagar, Gomes & Sillero (2023) suggested that microhabitat selection by species may be the most critical factor limiting their presence. This could explain why L. yanalcu is exclusively found in Site 3.

The presence of Liolaemus multicolor at Site 2, characterized by Andosol soil, may be explained by this species’ parental care behavior (Halloy et al., 2013) and/or acting against predators or for thermoregulation. The properties of Andosols likely facilitate burrowing (McDaniel et al., 2012). However, the relationship between burrowing performance and soil type remains poorly studied and has primarily been addressed in fossorial species (Ducey et al., 1993; Herrel & Measey, 2010; Doody et al., 2020; Barros et al., 2021). Several studies have documented the use of burrows by lizards, either for thermoregulation or as refuges from predators. For instance, Leiolepis belliana predominantly uses its burrows as an escape mechanism from predators rather than for thermoregulation, as internal burrow temperatures can exceed those of the external environment (Lei et al., 2021). Conversely, some lizard species rely on burrows constructed by other animals. For example, Liolaemus ruibali uses rodent burrows as refuges, which enhances its abundance (Bongiovanni, Borruel Díaz & Borghi, 2023), while Tiliqua adelaidensis utilizes spider burrows primarily for thermoregulation (Milne, Bull & Hutchinson, 2003). Additionally, Liopholis slateri builds burrows around the bases of shrubs and small trees, using them for various activities, including thermoregulation and predator avoidance (Fenner, Pavey & Bull, 2012). Similarly, Liolaemus multicolor excavates its own burrows at the base of Adesmia bushes, probably employing them for both thermoregulation and protection from predators. Moreover, Andosols retain higher levels of moisture and heat compared to Cambisols and Regosols (Lukito, Kouno & Ando, 1998; Cihacek & Bremner, 1988; Zhang et al., 2022). These thermal and hydric properties are crucial for offspring development (Deeming, 2004; Du & Shine, 2015), influencing growth, sexual maturation, and fitness (Wapstra et al., 2009; Li et al., 2018; Hao et al., 2021; Wang et al., 2021).

Our second hypothesis was partially refuted, since the present study did not find a relationship between microclimate variables and parity mode. Nevertheless, several studies have demonstrated that climatic variables, such as temperature and humidity, influence lizard reproductive strategies. Cold climates typically favor viviparity (Tinkle, Wilbur & Tilley, 1970; Hodges, 2004; Watson, Makowsky & Bagley, 2014; Ma et al., 2022, among others), which may limit oviparous species to lower distribution ranges. This hypothesis has been proposed for Liolaemus species (Fernández, Kubisch & Ibargüengoytía, 2017; Pincheira-Donoso et al., 2013; Cruz et al., 2014). Recently, Domínguez-Guerrero et al. (2024) found an association between viviparity and larger optimal body sizes, which could apply to L. albiceps, L. irregularis, and L. multicolor in the present study. However, L. yanalcu is the only oviparous species in this study (Martinez Oliver & Lobo, 2002; Quinteros, 2012; Abdala et al., 2021). The climatic conditions at Site 3 (high humidity and temperature) likely facilitate the occupancy of this species in a colder area, promoting egg development. Although there is evidence of some reproductive plasticity in oviparous lizards, including variation in incubation periods (Huey, 1977; Mathies & Andrews, 1996; Neill, 1964; Smith & Shine, 1997; Olsson et al., 2018), Ramirez-Pinilla (1992) found different egg retention times in Liolaemus species, including L. yanalcu. Microhabitat, along with microclimate and biological traits, play crucial roles in the distribution of L. yanalcu.

Liolaemus albiceps and L. irregularis exhibit several similar biological traits: both are viviparous, are of a similar size (~90 mm max SVL), and use shrubs as shelter (Espinoza, Wiens & Tracy, 2004; Abdala, 2007). The wind speed and humidity of Site 4 (being higher and lower, respectively, than those of Site 1) could explain the presence of L. albiceps at this site. The preferred temperatures measured by Valdecantos et al. (2013) do not differ between L. albiceps and L. irregularis at Site 4. These lizards likely select specific thermal microhabitats for thermoregulation, given the low predation risk at these high elevations in the Andes (Huey, Hertz & Sinervo, 2003; Valdecantos et al., 2013). It also seems that diet plays an important role in this pair of species; Liolaemus irregularis is omnivorous, while L. albiceps is herbivorous (Espinoza, Wiens & Tracy, 2004; Abdala et al., 2021). The microclimatic conditions of high humidity and lower temperature at Site 4 could influence the distribution of L. albiceps by increasing the presence of the plants this species eats. The extent of plant consumption is influenced by factors such as habitat type, with insular and arid environments favoring herbivory (Pietczak & Vieira, 2017). Additionally, as mentioned above, the study site exhibits low predation pressure, which may allow for prolonged digestion of plant material (Janzen, 1973; Van Damme, 1999). Furthermore, there are some documented cases of herbivores distribution affected by the plants they consume (Dilts et al., 2019; Carvajal Acosta & Mooney, 2021). Herbivorous lizards tend to inhabit areas where their preferred plant resources are readily available, as observed in Dicrodon guttulatum (Squeo et al., 2007; Van Leeuwen, Catenazzi & Holmgren, 2011), Phymaturus (Celedón-Neghme, Salgado & Victoriano, 2005; Castro, Laspiur & Acosta, 2013; Corbalán & Debandi, 2014), and even Liolaemus (Rocha, 2000). However, further research is needed to confirm the hypothesis that the availability of palatable plants significantly influences the distribution of lizard species.

The findings of this study contribute to a deeper understanding of how microhabitat variability shapes the ecological niches of Liolaemus species, providing a foundation for future studies on the impact of climate change on these and similar ectothermic species. The wide range of body temperatures and the diet of L. irregularis increases the survival ability of this species in diverse microhabitats, likely aiding its broad distribution. Conversely, L. yanalcu inhabits a site with higher humidity and temperature conditions, which supports its oviparous reproductive strategy, promoting egg development in otherwise colder climates. Liolaemus albiceps and Liolaemus multicolor prefer high humidity and low temperatures but differ in their soil use, with L. multicolor favoring Andosols, which may be related to the parental care behavior of this species. Moreover, we identified a relationship between microclimatic variables and certain biological traits of the species. We observed a distributional pattern driven by microclimatic variables, with no evidence that phylogenetic relationships influence species distribution across sites. Further studies that integrate detailed microclimate measurements, habitat use, and species interaction could provide deeper insights into these ecological dynamics.

Supplemental Information

Supplemental Information 1 Pos hoc probabilities of the comparisons of the sites between months made for the variables studied.

TEMP (Temperature); HUM (Relative Humidity); WS (Wind Speed); SR (Sun radiation). Values correspond to mean ± standard deviation. Letters at the right identify significate differences. Similar letters mean no difference.

Supplemental Information 2 Monthly variation of the variables studied.

A.- Monthly variation of Air Temperature of the four sites studied. Dots refers to mean and lines to standard deviation. B.- Monthly variation of Relative Humidity of the four sites studied. Dots refer to mean and lines to standard deviation. C.- Monthly variation of Wind Speed of the four sites studied. Dots represent the mean, and lines represent the standard deviation. D.- Monthly variation of Sun Radiation of the four sites studied. Dots refers to mean and lines to standard deviation.

Supplemental Information 3 Rasters of Air Temperature and Relative humidity.

Top. - Air Temperature raster, showing hourly variation on month of lizard’s main activity. Color dots identify the sites. Blue: Site 1; Red: Site 2; Green: Site 3; Orange: Site 4. Bottom. - Relative Humidity raster, showing hourly variation on month of lizard’s main activity. Color dots identify sites: Blue: Site 1; Red: Site 2; Green: Site 3; Orange: Site 4.

Supplemental Information 4 Code used for the microclimatic model.

Supplemental Information 5 Raw data of the microclimatic niche model.

We would like to thank E. Navarro, C. Abdala, J. M. Díaz Gómez, and F. Lobo for their assistance in the lab and their discussion of ideas related to this study. We would also like to thank A. Tálamo, A. Pietrek, and M. R. Ruiz-Monachesi for their help with the statistical analyses and J. J. Lauthier for help editing the manuscript. We are also grateful for the three anonymous reviewers who made comments and suggestions that greatly improved this manuscript.

Additional Information and Declarations

Competing Interests

The authors declare that they have no competing interests.

Author Contributions

Andrés S. Quinteros conceived and designed the experiments, performed the experiments, analyzed the data, prepared figures and/or tables, authored or reviewed drafts of the article, and approved the final draft.

Sabrina N. Portelli conceived and designed the experiments, performed the experiments, analyzed the data, prepared figures and/or tables, authored or reviewed drafts of the article, and approved the final draft.

Data Availability

The following information was supplied regarding data availability:

The raw data are available in the Supplemental Files.

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
