# Peer review of "The micro-niche explains allotopy and syntopy in South American Liolaemus (Iguania: Liolaemidae) lizards"

_PeerJ, doi:10.7717/peerj.18979_

## Round 0.1 · original submission · Major Revisions

Dear authors,

I appreciate the work that you have completed and the manuscript you have prepared for review. Considering the feedback from the three reviewers I have made a decision of "major revisions." All three reviewers agree that the work completed is an important contribution to the understanding of the impact of microniche differences on how fauna in general and reptiles/lizards in particular are distributed. Two reviewers, and myself, found that the manuscript as written is mostly descriptive and does not apply statistical analyses that are robust enough for establishing a causal relationship. I am in agreement with Reviewer #1 that additional analyses are needed for this manuscript to be considered for publication in PeerJ. When studying the reviews to decide a path to address these, I'd strongly suggest to look at Reviewer 1's suggestion of methodology to apply (e.g. include analyses relating microclimatic variables to the physiological and ecological characteristics of lizards, can use the approaches applied by Souza-Oliveira et al. (2024), etc).

Other suggestions that are common across the three reviewers relate to the English and grammar used, so I'd suggest a closer look at their suggestions in the comments and attached PDFs.

The above summary is not all encompassing, and I recommend you carefully analyze the feedback provided by all three reviewers.

Warm regards,
Carlos A. Santamaria

Reviewer 1 ·

Basic reporting

I have read the article " The Microclimatic Niche Explains Allopatry and Sympatry in South American Liolaemus (Iguania: Liolaemidae) Lizard (#104409)". While the premise of investigating the role of microclimatic variables in structuring lizard assemblages of Liolaemus in the South American Andes is compelling, I believe that the current analysis presents some limitations. Although the authors successfully characterize the microclimatic variables of each site, a clear connection (and statistical support) between these variables and species distribution is not established. The absence of statistical models that include species presence/absence as a response variable, as well as the lack of analyses relating microclimatic variables to the physiological and ecological characteristics of lizards (e.g., thermoregulation, diet), hinders a quantitative assessment of the impact of microenvironmental conditions on assemblage structure. In its current state, the manuscript is limited to a descriptive comparison of microclimatic conditions between sites, without delving into the mechanisms underlying species turnover. I suggest that new (or more) statistical analyses be incorporated, considering models that allow for the evaluation of the relationship between microenvironmental variables and species distribution, as well as the incorporation of information on species characteristics to better understand their response to environmental variations.

Experimental design

A more comprehensive statistical approach is needed to provide evidence supporting some of the claims made by the authors in the discussion section. Specifically:
i) In no statistical analysis was the presence/absence of species tested as a response variable as a function of explanatory microenvironmental variables. Therefore, the association between each species and the measured microenvironmental variables cannot be supported, and consequently, species turnover between sites and assemblages cannot be adequately explained.
ii) The association between the thermoregulatory parameters obtained from the literature (Valdecantos et al., 2013: average preferred temperature, Tset point of each species) and the microenvironmental variables measured in this study was also not analyzed. It should be noted that one of the studies cited by the authors of this manuscript is Souza-Oliveira et al. (2024) who in their methodology mention: "Multiple regressions were also performed to investigate which group of environmental variables (i.e., micro-climatic or canopy reflectance) best explain the compositional turnover of the lizards...". I suggest that the authors consider an analogous approach.
iii) While the introduction and discussion frequently highlight the phylogenetic relationships among Liolaemus species within each assemblage, the manuscript lacks a statistical assessment of phylogeny's contribution to species distribution. Consequently, it remains unclear whether phylogenetic factors or microenvironmental variables primarily influence species distribution and assemblage structure.

Validity of the findings

The excessive number of isolated figures hinders the visualization of contrasts between sites in a global manner for the set of environmental variables. It is suggested to incorporate a multivariate analysis (e.g., PCA or DFA) to generate a figure that summarizes the segregation of sites based on multiple variables.
Supplementary Table 1 presents inconsistencies in nomenclature (English/Spanish) and in the format of the numerical data (number of decimal places). Additionally, its content is more relevant than the table 4, so it is recommended to interchange their positions.
At line 213, where it states: “the structure of Liolaemus species communities...” This statement about the role of microclimate in structuring Liolaemus assemblages lacks empirical support. The authors should reconsider this statement in light of the following limitations:
i) There is a lack of analyses that relate the presence/absence of species to microenvironmental variables, preventing the establishment of a causal relationship. ii) The thermoregulatory parameters of Valdecantos et al. (2013) have not been integrated into the statistical analyses, despite their relevance for understanding the relationship between species and the microenvironmental variables measured in this study. iii) The influence of phylogeny on species distribution has not been statistically evaluated, which prevents determining whether the structure of assemblages is a product of evolutionary history or current environmental conditions.

At lines 251-256, where it states "Adequate soil conditions, which maintain temperature and humidity, likely contribute to efficient juvenile thermoregulation. Furthermore, offspring of lizards exposed to warmer environments have shown better survival rates under simulated warming conditions than under current climate scenarios, suggesting that parental exposure to specific soil temperatures can induce adaptive changes in offspring, enhancing their survival prospects in changing environments (Sun et al., 2018)." This section on juvenile thermoregulation is speculative and lacks empirical support in the present study. The statements about the influence of soil on juvenile thermoregulation and the effects of environmental conditions on offspring survival are not supported by the presented data.
At L260-262, where it stated "Therefore, the higher temperatures at Site 3 allow L. yanalcu to regulate its body temperature more effectively. High relative humidityat Site 3 also reduces the rate of evaporative water loss in L. yanalcu." The present dataset does not provide sufficient evidence to substantiate this assertion. To establish a more robust conclusion, additional statistical analyses are required to explore the relationships between thermoregulatory parameters, as measured in previous studies (e.g., Valdecantos et al., 2013), and the microclimatic variables examined herein.

Additional comments

INTRODUCTION
At L102 where it stated “These four species share and differ in various biological traits.” This statement is broad and imprecise. To which specific biological characteristics does it refer?
At L106-108 where it stated "Using these data, we characterized the microclimate of each site and, based on this characterization, explained the potential reasons behind the distribution of these four species. Additionally, we correlated the microclimate data with biological traits such as parity mode, diet, and thermoregulation skills of the species." While the statistical analyses effectively characterize and contrast climatic variables between sites, further investigation is required to fully understand the relationship between these variables and Liolaemus assemblage composition. As outlined in my previous comments, additional analyses are necessary to address the study's broader objectives.

Reviewer 2 ·

Basic reporting

The manuscript still contains quite a few errors in the writing

Experimental design

"The study employed models with input data on microclimates and other environmental parameters to compare ecological niches of four different regions, in which four Liolaemus species inhabit sympatrically or allopatrically. I highly appreciate the scientific value of the research.

Validity of the findings

The discussion section was very well written and explaining why the species are distributed in their region based on the correlation with microclimate and habitat. However, the manuscript should consider some suggestions I have noted. I fully agree that the manuscript is suitable for publication in the journal after addressing the comments mentioned in PDF files of review.

Additional comments

"The study employed models with input data on microclimates and other environmental parameters to compare ecological niches of four different regions, in which four Liolaemus species inhabit sympatrically or allopatrically. I highly appreciate the scientific value of the research. The discussion section was very well written and explains why the species are distributed in their region based on the correlation with microclimate and habitat. However, the manuscript should consider some suggestions I have noted below. I fully agree that the manuscript is suitable for publication in the journal after addressing the comments mentioned below and the PDF file of review:
The title of manuscript: “The Microclimatic niche”, but models you employed variable of soil type and data on elevation. I would suggest you rename: “The micro-environmental niche” or “The mico-niche”
Line 29-30: the sentence should be revised and add which microclimates the model used. The model you used, did it employ the variable of soil type? If yes, the model should be the environmental model, not microclimatic model.
Line 30-34: You should provide a summary sentence the characteristics among four study sites to see whether they are different or similar to each other." And revise the comparison, it seems that only site 4, when you used “while”, differs to three remaining sites.
Line 35: please revise: “an emerging area of research”???
Line 73: which data?
Line 106-107: Revising the sentence and clearly state the objective
Line 124-126: it’s an aim of research, not a method.
Line 112 and Line 127-134: The part should only be the study area. And introduction of phylogenetic relationships among target species should be presented in the introduction. You should only present how may occurrences of each species did you use for the micro-niche model
Line 171 – 173: should be in the method and revise the first sentence of result
Line 178 – 180: Summarizing into one sentence
Line 182: please revise: “Conversely, Sites 2 and 4 show similar variations, differing primarily 183 in soil type”

Annotated reviews are not available for download in order to protect the identity of reviewers who chose to remain anonymous.

Reviewer 3 ·

Basic reporting

The results presented in the paper are interesting as not much has been reported on the analysis of lizard microhabitats, and it is therefore a contribution to the state of knowledge on lizard habitat use at the micro level.
The English used is not of the best quality, with some grammatical errors and in some parts difficult to follow and understand the main point. I believe that the authors could easily correct this if the paper was checked by a native speaker or other grammatical aids were available.
I think that Figures 2, 3 and 4 could be made into a composite figure. Figures 6 and 7 are redundant. Table 1 is redundant (repeats the plots in Figures 2-4). Table 2 can be in the Supplementary.

Experimental design

The investigation of microhabitat characteristics using a modelling approach and retrieval of relevant data from different datasets was well planned and executed. The aim was to compare the microhabitat characteristics of four different sites harbouring either one or two different lizard species. The results obtained using microhabitat modelling were accurately compared. The methodology is well explained.

Validity of the findings

The study has failed to test some relevant hypotheses because the paper is not well structured. The introduction gives a very broad background on species distribution modelling (which was not the focus of this study) and a limited background on microhabitat studies on lizards (with almost no relevant literature), and lacks background on the study system (this is then explained in the discussion, but I would suggest moving this to the introduction and using it to make predictions about their microhabitat at sites where they live together or not from the large body of knowledge on these species. Since the study uses the terms "sympatry" and "allopatry" (really it should be "syntopy" and "allotopy" if the site is a small area inhabited by both species), I was eager to read about interspecific interactions and habitat use, but the part about interspecific interactions is missing from the paper. Do these species interact in any way and how? Would you expect microhabitat differences between sites to help promote coexistence at some sites and not others, and why? Finally, the conclusions are very general as the study only compares a single site with a single species pair and does not include replicates, which would make the results more robust and allow such conclusions to be drawn. This can be improved by referring to the analysis at a single site in the discussion and avoiding general conclusions.

Additional comments

Overall, the study has potential, but the main comment is that a testable hypothesis should be developed in the introduction based on the known biology of these species.

---

## Round 0.2 · Minor Revisions

I appreciate the work that you and your co-authors have taken to address the concerns and feedback provided by the reviewers. I am returning a minor revisions decision to give you an opportunity to address the feedback from Reviewer 1.

Reviewer 1 ·

Basic reporting

Dear Author
I have read the manuscript entitled “The micro-niche explains allotopy and syntopy in
South American Liolaemus (Iguania: Liolaemidae) lizards”. Compared to the previous version, I have observed a substantial improvement in the overall quality of the manuscript. However, I have some queries:

L116-122. Before presenting the hypotheses, it would be advisable for the author to state the general objective and the specific objectives of the study. For example:

The main objective was to determine the factors influencing the assembling pattern (allotopy or syntopy) of four Liolaemus species (L. albiceps, L. irregularis, L. multicolor, L. yanalcu) in northwestern Argentina, considering both microenvironmental variables and intrinsic species characteristics. The specific objectives were:
(1) To compare microclimatic conditions across four sites.
(2) To analyze the relationship between species presence/absence and microenvironmental variables and biological characteristics.
(3) To investigate the association between microclimatic variables and species biological characteristics.
(4) To explore the influence of phylogeny on spatial distribution and assembly patterns.

The Discussion section requires significant revision. Its current organization hinders the logical flow of arguments. To enhance clarity and coherence, I propose reorganizing the section to align with the specific objectives and hypotheses outlined in the Introduction. This approach will facilitate a more focused discussion and enable a clearer evaluation of the support or rejection of each hypothesis. Particular emphasis should be placed on explicitly stating which hypotheses are accepted and which are rejected.

Experimental design

The experimental design and analytical approach are appropriate.

Validity of the findings

The Discussion section requires significant revision. Its current organization hinders the logical flow of arguments. To enhance clarity and coherence, I propose reorganizing the section to align with the specific objectives and hypotheses outlined in the Introduction. This approach will facilitate a more focused discussion and enable a clearer evaluation of the support or rejection of each hypothesis. Particular emphasis should be placed on explicitly stating which hypotheses are accepted and which are rejected.


L261-263. The information presented in this paragraph does not discuss the findings of the present study. As written, it seems more appropriate for the introduction section. This information needs to be linked to the results of the current study.

L264-272. This paragraph should be placed at the beginning of the discussion.

L271. This study did not investigate microhabitat preferences but rather microhabitat use. The term 'preference' was avoided and replaced with 'use'

L282-285. "Studies have shown that smaller lizards lose body heat more quickly than larger ones (Sagonas et al., 2013; Moreno et al., 2016, Claunch et al., 2021). Therefore, the higher temperatures at Site 3 allow L. yanalcu to regulate its body temperature more effectively."

Small-sized organisms exhibit low thermal inertia, rendering them more susceptible to fluctuations in body temperature. This limited capacity to store heat may compromise the efficiency of their thermoregulatory mechanisms. It is crucial to consider that heat exchange between an organism and its environment is a dynamic and bidirectional process, not a unidirectional flow as the original statement suggests.

L286-287. "High relative humidity at Site 3 probably reduces the rate of evaporative water loss in L. yanalcu."

This statement is not fully addressed in the paragraph. There is a relationship between hydroregulation and thermoregulation in species. This point should be further elaborated.

See:
Rozen‐Rechels, D., Dupoué, A., Lourdais, O., Chamaillé‐Jammes, S., Meylan, S., Clobert, J., & Le Galliard, J. F. (2019). When water interacts with temperature: Ecological and evolutionary implications of thermo‐hydroregulation in terrestrial ectotherms. Ecology and evolution, 9(17), 10029-10043.
Trewartha, D. M., Clayton, J. L., Godfrey, S. S., & Gardner, M. G. (2024). Heat water and reptiles–do the hydro‐thermal properties of animals at the source location persist at the translocation site?. Animal Conservation.

L291-292. "Additionally, Site 3 features some Festuca sp. grasslands, which serve as shelter for L. yanalcu (Martinez Oliver & Lobo)"

How do shrubs provide refuge for lizards? Do they serve as protection against predators, as thermally favorable microhabitats, or both?
See

Žagar, A., Gomes, V., & Sillero, N. (2023). Selected microhabitat and surface temperatures of two sympatric lizard species. Acta Oecologica, 118, 103887.
Stellatelli, O. A., Vega, L. E., Block, C., & Cruz, F. B. (2013). Effects of tree invasion on the habitat use of sand lizards. Herpetologica, 69(4), 455-465.

L293-298. "The presence of Liolaemus multicolor at Site 2 with Andosol soil type could be explained by this species’ parental care behavior (Halloy et al., 2013)"

Further discussion is needed at this point, incorporating scientific evidence to support the claims made. It is suggested to delve deeper into the following questions: What specific physical properties make this type of soil more suitable for lizards? Are there previous studies supporting the hypothesis that this soil facilitates burrow excavation and improves offspring survival?

L299-315. "Although the present study did not find a relationship between microclimate variables and parity mode, several studies have demonstrated that climatic variables, such as temperature and humidity, influence lizard reproductive strategies..."

This paragraph discusses trends that are not supported by the data from the present study. I suggest focusing the discussion on the obtained findings.

Additional comments

no comments

---

## Round 0.3 · Minor Revisions

Thank you for the additional changes that you have made to your manuscript. The improvements are noted by both Reviewer #1 and myself. Most of the changes required at this point are to the writing in the manuscript which should be straight forward to address and would bring the manuscript to publication quality if addressed correctly. Please address the feedback provided by Reviewer #1.

Reviewer 1 ·

Basic reporting

I have evaluated the revised version of the manuscript titled “The micro-niche explains allotopy and syntopy in South American Liolaemus (Iguania: Liolaemidae) lizards (#104409)” and consider it to have undergone a significant improvement compared to previous versions. The authors have satisfactorily addressed the comments made in my earlier reviews, particularly those related to the formulation of the objectives. However, the conclusions section requires some minor adjustments, such as the elimination of redundancies, a more in-depth discussion of certain findings, and greater bibliographic support. Below, I present details of my observations.
L142-147. " This study analyzed microclimatic variables and soil types obtained from Niche MapR (Kearney & Porter, 2017) and SoilGrids (Hengl et al., 2017). To test the hypotheses, the microenvironmental niche of each site was characterized and the microclimate data was correlated with biological traits of the species, such as parity mode, diet, and thermoregulatory abilities. This study proposes potential explanations for the distribution patterns of the four species based on the results of these analyses."
The final sentence in the introduction is redundant and could be omitted. Some of the information it presents, especially regarding variables, should be incorporated in more detail into the specific objectives and hypotheses. Additionally, certain methodological elements are misplaced in this section and should be moved to the Methodology section.
L269. " Our results support our hypotheses."
The authors need to clarify if this generalization pertains to all the hypotheses presented. Nevertheless, I contend that each hypothesis necessitates a thorough and independent examination.
L271. " Finally,"
The term 'finally' should be deleted. Moreover, certain content within this paragraph seems to prematurely anticipate the conclusion and would be better positioned in the final paragraph.
L274. " Despite the homogeneity of the landscape of the four study site...."
The reference to 'landscape homogeneity' is ambiguous. Does it refer to homogeneity in terms of structure, such as vegetation structure or relief? Given that the authors later mention changes in the substrate, it would be advisable to reconsider or clarify this term.
L274-282. The concise information presented in this paragraph has the potential to substantially enhance the opening discussion and provide a valuable complement to the first paragraph (L261-273). To facilitate a more thorough and timely exploration of the implications of these crucial findings, it is suggested that they be repositioned at the beginning of the discussion section.
L284. " significant plasticity"
The species in question exhibits a broad tolerance to varying thermal and structural conditions, suggesting a relatively generalist nature. The obtained results align with a generalist profile but do not support the hypothesis of high plasticity.
L287. "Liolaemus" Italicize this word.
L307. " L. yanalcu" Italicize these words.
L308-311. " This shelter likely serves two functions: providing protection against predators, as lizards inside the shelter are difficult to detect, and acting as a thermally favorable microhabitat, as individuals have been observed basking outside the grassland (our unpublished data)."
Vegetation plays a crucial role in the ecology of lizards, including Liolaemus, by providing both refuge from predators and suitable thermal microhabitats. Numerous studies support these claims. See:
Smith, G. R., & Ballinger, R. E. (2001). The ecological consequences of habitat and microhabitat use in lizards:: a review. Contemporary Herpetology, 1-28.
Monasterio, C., Salvador, A., Iraeta, P., & Díaz, J. A. (2009). The effects of thermal biology and refuge availability on the restricted distribution of an alpine lizard. Journal of Biogeography, 36(9), 1673-1684.
Herczeg, G., Herrero, A., Saarikivi, J., Gonda, A., Jäntti, M., & Merilä, J. (2008). Experimental support for the cost–benefit model of lizard thermoregulation: the effects of predation risk and food supply. Oecologia, 155, 1-10.
Stellatelli, O. A., Vega, L. E., Block, C., & Cruz, F. B. (2013). Effects of tree invasion on the habitat use of sand lizards. Herpetologica, 69(4), 455-465.
L318. " The properties of Andosols likely facilitate burrowing"
The authors suggest that the ease of substrate excavation could increase the availability of refuges for lizards, especially during reproduction. While this idea is intriguing, the presented data do not establish a clear causal relationship with reproduction. However, it would be interesting to discuss whether the availability of burrows or the presence of a potentially excavatable substrate could influence the presence of lizards, which use burrows as refuges from predators and offer favorable microclimatic conditions that contrast with the external environment at certain times of the day and year. There are studies that support these claims, see:
Lei, J., Binti Yusof, N. S., Wu, N. C., Zhang, Z., & Booth, D. T. (2021). The burrowing ecology of a tropical lizard (Leiolepis belliana). Herpetologica, 77(1), 37-44.
Bongiovanni, S. B., Borruel Díaz, N. G., & Borghi, C. E. (2023). Ecosystem engineering in a high and cold desert: Effect of a subterranean rodent on lizard abundance and behaviour. Austral Ecology, 48(7), 1238-1244.
L348-354. " These lizards likely select specific thermal microhabitats for thermoregulation, given the low predation risk at these high elevations in the Andes (Huey, Hertz & Sinervo, 2003; Valdecantos et al., 2013). It also seems that diet plays an important role in this pair of species; Liolaemus irregularis is omnivorous, while L. albiceps is herbivorous (Espinoza et al., 2004; Abdala et al., 2021). The microclimatic conditions of high humidity and lower temperature at Site 4 could influence the distribution of L. albiceps by increasing the presence of the plants this species eats"
The paragraph provides a concise summary of the findings, especially concerning the hypothesis that diet is the primary factor driving niche segregation and coexistence between the two species. Nevertheless, a more thorough discussion, backed by relevant citations, is needed to fully support this claim. The authors could enhance their argument by citing prior research that corroborates this hypothesis and by delving deeper into the underlying ecological mechanisms.L355-364. " The findings of this study contribute to a...species."; L365-370. " This study underscores the importance....dynamics"
The last two paragraphs, although they effectively summarize the principal findings and their implications, contain some redundancy with the content introduced in the opening paragraph of the discussion section (lines 261-273). I suggest merging these paragraphs and restructuring them into a concise conclusion, incorporating elements of the initial paragraph as needed.

Experimental design

no comment

Validity of the findings

no comment

---

## Round 0.4 · accepted · Accept

Thank you for taking the time to address all comments provided by the reviewers throughout the multiple rounds of review we have undertaken. After reviewing your latest draft, it is my opinion that you have addressed all the concerns and feedback that the reviewers have brought up. Thus, I am happy to render an "Accept" decision.

I did notice three small issues that need to be fixed during the production process. They are the following:

Line 91: space between "far," and "no"
Line 297: space between "a" and "generalist"
Line 406: space between "species." and "The"

Congratulations!